# Transcriptomic Profiling Reveals AKR1C1 and AKR1C3 Mediate Cisplatin Resistance in Signet Ring Cell Gastric Carcinoma via Autophagic Cell Death

**DOI:** 10.3390/ijms222212512

**Published:** 2021-11-19

**Authors:** Nang Lae Lae Phoo, Pornngarm Dejkriengkraikul, Patompong Khaw-On, Supachai Yodkeeree

**Affiliations:** 1Department of Biochemistry, Faculty of Medicine, Chiang Mai University, Chiang Mai 50200, Thailand; nanglaelae_phoo@cmu.ac.th (N.L.L.P.); pornngarm.d@cmu.ac.th (P.D.); patompong_k@cmu.ac.th (P.K.-O.); 2Center for Research and Development of Natural Products for Health, Chiang Mai University, Chiang Mai 50200, Thailand

**Keywords:** signet ring cell gastric carcinoma, RNA sequencing, aldoketoreductase, autophagy, drug resistance

## Abstract

Signet ring cell gastric carcinoma (SRCGC) is a lethal malignancy that has developed drug resistance to cisplatin therapies. The aim of this study was to characterize the acquisition of the cisplatin-resistance SRCGC cell line (KATO/DDP cells) and to understand the molecular mechanisms underlying cisplatin resistance. Transcriptomic and bioinformatic analyses were used to identify the candidate gene. This was confirmed by qPCR and Western blot. Aldoketoreductase1C1 and 1C3 (AKR1C1 and AKR1C3) were the most promising molecules in KATO/DDP cells. A specific inhibitor of AKR1C1 (5PBSA) and AKR1C3 (ASP9521) was used to enhance cisplatin-induced KATO/DPP cell death. Although cisplatin alone induced KATO/DDP apoptosis, a combination treatment of cisplatin and the AKR1C inhibitors had no influence on percent cell apoptosis. In conjunction with the autophagy inhibitor, 3MA, attenuated the effects of 5PBSA or ASP9521 to enhance cisplatin-induced cell death. These results indicated that AKR1C1 and 1C3 regulated cisplatin-induced KATO/DDP cell death via autophagy. Moreover, cisplatin in combination with AKR1C inhibitors and N-acetyl cysteine increased KATO/DDP cells’ viability when compared with a combination treatment of cisplatin and the inhibitors. Taken together, our results suggested that AKR1C1 and 1C3 play a crucial role in cisplatin resistance of SRCGC by regulating redox-dependent autophagy.

## 1. Introduction

Gastric cancer is the fifth most common threat to human health worldwide according to GLOBOCAN 2018. It has been associated with a significantly high mortality rate, making it now the third leading cause of cancer-related deaths [1]. In the last few years, there has been an overall decline in incidences of cancer; however, some studies have reported that there has been an abrupt increase in the occurrence of signet ring cell gastric cancer (SRCGC) in Asia, Europe, and the United States [2]. In recent years, effective treatments for gastric cancer have included surgery, chemotherapy, and targeted therapy. However, the degree of efficacy of these forms of treatment has not yet reached expectations. Platinum drugs (cisplatin and oxaliplatin) appear to be effective as a first-line treatment in SRCGC patients [3], although many patients still suffer from relapses and may eventually become resistant to chemotherapy. This has contributed to a decrease in the 5-year survival rate. At present, many researchers are now focusing on various mechanisms to reverse the cisplatin resistance of SRCGC.

Cisplatin, once it enters the cell; becomes activated and binds to DNA. The platinum atom of cisplatin binds to the N7 atom of purine nucleotide and creates intrastrand and interstrand crosslinks and disturbs the structure of DNA, which further stimulates the DNA repair process and apoptosis cascades [4]. Cisplatin not only attacks nuclear DNA, but also attacks mitochondrial DNA and induces mitochondrial reactive oxygen species (ROS) production and promotes cell death [5]. Excessive reactive oxygen species can stimulate not only apoptosis via both the extrinsic and intrinsic pathways, but can also induce autophagy, which is a self-catabolic process that induces the sequestration of cytoplasmic contents, including exhausted organelles and protein aggregates for degradation in lysosomes [6].

The poor response of gastric cancer to cisplatin chemotherapy is usually due to combined mechanisms of chemoresistance, which may include a reduction in drug uptake, enhanced drug efflux, and a reduction in the proportion of active agents in tumor cells due to a reduction in pro-drug activation or an enhancement in drug inactivation. Other mechanisms of chemoresistance include changes in the expression or function of the molecular targets of anticancer drugs, the enhanced ability of cancer cells to repair anticancer drug-induced DNA damage, and a decrease in the expression of proapoptotic factors or an upregulation in antiapoptotic genes [7]. In addition, recent studies have shown that overexpression of glutathione transferase P1-1, UGT1A1, microsomal cytochrome P450 enzymes (CYPs), and aldoketoreductases (AKRs) were involved in development of cisplatin resistance in various types of cancer by neutralizing the cytotoxicity of the drug [8,9,10,11]. Presently, there exists a lack of published studies on how cisplatin resistance has been developed in signet ring cell gastric carcinoma patients, as they have been known to possess distinctive pathological features from other cell types of gastric cancer.

Nowadays, transcriptomic sequencing technologies are considered very practical and beneficial revolutionary tools in genomic cancer research, drug resistance, and determining the prognosis of various forms of cancer [12]. Therefore, we used this technique to comprehensively analyze the differences in gene expression between cisplatin resistance in SRCGC cells and their parental cells. Such transcriptome profiling will help to build a solid foundation for further in-depth studies of the mechanisms of cisplatin resistance in SRCGC cells.

In order to explore the underlying mechanisms of cisplatin-resistant SRCGC cells, we developed a cisplatin-resistant gastric cancer cell line, namely KATO/DDP, from its parental cell line KATOIII and compared the relevant gene expression profiles using the transcriptome sequencing technique. Bioinformatic analysis revealed that the oxidation reduction pathway, the steroid hormone metabolic process, and the small molecule metabolic process were all more active in KATO/DDP cells. All of these pathways indicated the presence of aldoketoreductase 1C1 and 1C3 genes (*AKR1C1* and *1C3*). These highly expressed genes were further confirmed by qPCR and Western blot analysis. Furthermore, we effectively inhibited AKR1C1 and AKR1C3 activity-induced cell death via autophagy. AKR1C1 and 1C3 have already been implicated in therapeutic resistance via the mediation of intracellular ROS levels for several types of cancer [13,14]. By using the inhibitors of AKR1C1 and AKR1C3, our results confirmed that high levels of AKR1C1 and 1C3 in KATO/DDP could reduce cisplatin-induced cell death via regulated intracellular ROS. Thus, our findings indicated that AKR1C1 and AKR1C3 play an important role in cisplatin-resistant SRCGC via the regulation of autophagic cell death. 

## 2. Results

### 2.1. Long-Term Exposure to Cisplatin Treatment Can Promote Drug Resistance in SRCGC 

The SRCGC cells (KATOIII cells) were treated with a stepwise concentration of cisplatin at 0.5 to 3 µM over 9 months to generate acquired resistance cells. This resistant cell line was termed KATO/DDP. Cell viability was assessed using a trypan blue cell exclusion assay. As shown in Figure 1A,B, the IC_50_ of KATOIII was found to be 12.67 ± 1.155 µM and of KATO/DDP was reported to be 98 ± 1.732 µM, while the resistance ratio was 7.7.

### 2.2. Identification of Candidate Genes to Promote Cisplatin Resistance in SRCGC Using Transcriptomic Sequencing

An RNA-seq approach was used to assess differential expression across the transcriptome in cisplatin-resistant SRCGC cells, KATO/DDP, and its parental cell, KATOIII. The results indicated that there were 519 differentially expressed protein coding genes, with 174 upregulated and 345 downregulated genes when log2 fold change values were ≥1.8 and ≤−1.8 (Figure 2A). In order to identify the key deregulated genes, the top 50 upregulated and downregulated genes (fold change) were determined for KATO/DDP versus KATOIII cells. The top 50 upregulated and downregulated genes are shown in Appendix A. We then detected the expression of five randomly selected upregulated and five downregulated genes by qPCR to validate the results of the RNA sequencing (Figure 2B). The qPCR results revealed an upregulation of genes *ALDOB, HSD17B2, AKR1C1, PLAC8,* and *AKR1C3,* and the downregulation of genes *FGF21, CPXM2, TBC1D4, FREM2,* and *ALDH1L2,* showing the same pattern with transcriptomic results. To determine whether cisplatin resistance in KATO/DDP cells may be related to a certain biological pathway, a GO functional enrichment analysis was conducted. In GO annotations, all enriched differentially expressed genes (DEGs) were classified into three categories: biological processes, cellular components, and molecular functions. Results of the top 10 significant pathways in biology processes are shown in Figure 3A, while the enriched genes of each pathway are listed in Appendix A. Interestingly, the AKR gene family has been associated with the topmost significant pathways, including those associated with oxidation reduction, digestion, and the steroid hormone metabolic process, as well as the doxorubincin and daunorubicin metabolic processes.

Moreover, the Gene Set Enrichment Analysis results confirmed that the oxidation reduction pathway, the small molecule metabolic process, and the cofactor metabolic process all belonged to the topmost significant pathways with the highest enrichment scores. All these pathways indicated the presence of the AKR family with higher ranking matrix scores (Figure 3B–E). Genes involved in oxidation reduction pathway, the small molecule metabolic process, and the cofactor metabolic process are listed in Appendix A. The protein–protein interaction network was constructed from differentially expressed genes using Cytoscape version 3.8.2, wherein 371 nodes with 483 edges were identified. From this network, the most significant module was detected using the MCODE Cytoscape plugin. MCODE revealed 13 nodes and 36 edges, with the highest score 6 containing *UGT1A1, UGT1A10, CYP1A1, CBR3, HSD17B2, AKR1C1, AKR1C3, CXCL8, CXCL11, CXCL13, NMUR2, ADCY7,* and *SSTR5* (Figure 4A,B). All these combined bioinformatic results indicated that AKR1C1 and AKR1C3 could be the most likely molecular markers to promote cisplatin resistance in the KATO/DDP cell line.

### 2.3. Verification of Cisplatin Resistance Related Genes in KATO/DDP Cell Line

Previous studies have reported that cisplatin resistance was associated with the impairment in drug transporters and DNA repair proteins. Consequently, we first determined the expression levels of major DNA repair genes such as *XRCC1* and *ERCC1*, and drug transporters such as *MRP1* and *MDR1*, which are known to be associated with drug resistance. The results indicated that there were no differences in expression values between the KATOIII and KATO/DDP cell lines in qPCR (Figure 5A). According to previously published reports, AKR family members are considered important genes in promoting drug resistance in the KATO/DDP cell line. The expression levels of *AKR1C1, 1C2, 1C3, 1B1,* and *1B10* genes were compared to those of KATOIII and KATO/DDP using qPCR. As shown in Figure 5B, AKR1C3 and AKR1C1 exhibited the highest fold-change difference between KATOIII and KATO/DDP at 6.2- and 5.6-fold, respectively, while the fold-change difference of AKR1C2 was 2.2. The expression levels of AKR1B1 and 1B10 indicated that there were no differences between KATOIII and KATO/DDP. This suggested that AKR1C1 and 1C3 were dramatically overexpressed in resistance phenotype KATO/DDP when compared to sensitive phenotype KATOIII cells. Moreover, AKR1C3 exhibited the highest expression level among members of the AKR family in KATO/DDP using qPCR (Figure 5C). In the subsequent step, we attempted to endorse differences in the protein expression levels of AKR1C1 and AKR1C3 between the KATOIII and KATO/DDP cell lines using Western blot analysis. As shown in Figure 5D–F, the expression levels of AKR1C1 and 1C3 were increased by cisplatin treatment in KATO/DDP cells, whereas the expression levels of both proteins in KATOIII cells after treatment with cisplatin did not change. All these combined results indicated that AKR1C1 and AKR1C3 are the most presiding molecules in promoting cisplatin resistance in the KATO/DDP cell line.

### 2.4. Inhibition of AKR1C1 and AKR1C3 in Resensitizing the Cisplatin Cytotoxicity 

In order to verify that AKR1C1 and AKR1C3 were the most promising factors in promoting cisplatin resistance in SRCGC, AKR1C1 and AKR1C3 were specifically inhibited by 3-bromo-5-phenylsalicylic acid (5PBSA) and ASP9521, respectively. KATO/DDP cells were treated with different concentrations of cisplatin (0–100 µM) with or without an AKR1C1 or AKR1C3 inhibitor. When KATO/DDP cells were treated with cisplatin at 25 µM together with ASP9521 at 5 and 10 µM, there was a significant reduction in cell viability, from 77 ± 1.1% to 48 ± 1.0% and 45 ± 2.0%, respectively. The same results were also found in KATO/DDP cells that had been treated with cisplatin together with 5PBSA at 10 µM (Figure 6A,B). Cisplatin often exerts cytotoxicity by inducing apoptosis in various tumor cells, including those associated with gastric cancer. Next, we investigated whether the enhancement activity of AKR1C inhibitors on cisplatin-induced cell death was associated with apoptosis by employing an annexin V-PI staining assay. The results indicated that treatment with cisplatin alone at 25 µM induced apoptosis population in KATO/DDP cells from 7.6% to 13%. However, cotreatment with ASP9521 or 5PBSA at 10 µM, and cisplatin at 25 µM, induced the apoptotic population to a degree that was similar to the treatment with cisplatin alone (Figure 6C,D). To confirm whether apoptosis was the main cause of AKR1C inhibitors enhancing cisplatin-induced cell death, levels of the apoptotic signaling pathway proteins were investigated by including cleaved caspase-3 and Bax. As shown in Figure 6E, a combination treatment did not change the level of cleaved caspase-3 and Bax when compared with cisplatin alone. Although there was a significant reduction in the percentage of cell viability in combination treatment of cisplatin and AKR1C inhibitors when compared to cisplatin alone, no significant changes were observed in apoptosis assay and apoptosis markers between cisplatin alone and cisplatin treated together with AKR1C inhibitors. The data suggested that the cell death induced by the inhibition of AKR1C1 and AKR1C3 was independent of the apoptosis pathway, and utilized an alternative pathway to promote cell death.

### 2.5. Inhibition of AKR1C1- and AKR1C3-Induced Cell Death via Autophagy 

Cisplatin-induced cell death occurred via the apoptosis pathway, but also stimulated autophagy cell death. Therefore, we investigated whether the enhancement activity of the AKR1C1 and 1C3 inhibitors on cisplatin-induced cell death was involved with autophagy. Autophagy vacuoles were labeled by monodansylcadaverine (MDC) fluorescent staining and detected using a fluorescent microscope. As shown in Figure 7A,B, the formation of autophagic vacuoles in KATO/DDP cells for the combined treatment of cisplatin and ASP9521 or 5PBSA were increased from 33 ± 1.3% to 51 ± 3.3% and 52 ± 1.2%, respectively, when compared to cisplatin alone. To further confirm that AKR1C1 and 1C3 inhibitors mediated cisplatin-induced KATO/DDP cell death via the autophagy pathway, the expression level of LC3B-II, a credible marker of autophagosome, was assayed by Western blot analysis. As shown in Figure 7C,D, there was an increase in the expression of LC3B-II in combination treatment with cisplatin and ASP9521 or 5PBSA when compared to the treatment with cisplatin alone. 

To verify that autophagy played a major role in the process wherein AKR1C1 and 1C3 inhibitors enhanced cisplatin-induced cell death, KATO/DDP cells were cotreated with 3MA (autophagy inhibitor), cisplatin, and AKR1C1 or 1C3 inhibitors for 48 h. Subsequently, the degree of cell viability was analyzed. As shown in Figure 8A,B, cell survival was increased by up to 66 ± 2.3% when cells were treated with 3MA together with ASP9521 and cisplatin. In addition, cell survival also increased by up to 68 ± 1.4% in cells treated with 3 methyl adenine (3MA) together with 5PBSA and cisplatin, according to the pattern observed in the resistance phenotype. Furthermore, we assessed the outcome of 3MA by using MDC autophagy vacuole staining. We confirmed that there was a reduction in the autophagic vacuole to 18 ± 0.56% and 17 ± 0.85% in cells treated with 3MA in conjunction with cisplatin and ASP9521 or 5PBSA when compared to cisplatin in conjunction with ASP9521 or 5PBSA (Figure 8C,D). Taken together, inhibition of AKR1C1 and 1C3 increased the formation of autophagic vacuoles, as well as LC3B-II formation and promotion of cell death. This suggested that AKR1C1 and 1C3 inhibition regulated the cisplatin-induced KATO/DDP cell death via the autophagy pathway.

### 2.6. AKR1C1 and 1C3 Mediated Chemo-Resistance in KATO/DPP by Regulating Redox Homeostasis

Recent reports have postulated that cisplatin increased the generation of intracellular ROS, which then caused damage to DNA, proteins, and lipids, leading to cell death. Human AKR1C isoforms were involved in the reduction of 4-hydroxynonenal (HNE), a reactive aldehyde derived from lipid peroxidation, into its less-toxic alcohol 4-hydroxy-2-nonenol form. To further clarify the underlying mechanisms of AKR1C1- and 1C3-mediated cisplatin-resistance in SRCGC, intracellular ROS levels were determined using a DCF-DA ROS assay. As shown in Figure 9A, there was a significant increase in intracellular ROS after being treated for 1 h with cisplatin at 25 µM in KATO III cells. In contrast, there were no significant changes in AKR1C1 and AKR1C3 overexpressing KATO/DDP. Moreover, when KATOIII cells were treated together with cisplatin and N-acetyl cysteine (NAC) 2 mM, cell viability increased back to 65 ± 2.1% from 46 ± 1.0% when compared to treatment with cisplatin alone (Figure 9B). These results indicated that cisplatin induced KATO III cell death, at least on the part of intracellular ROS generation. To further investigate the underlying mechanisms of AKR1C1- and AKR1C3-mediated cisplatin resistance in KATO/DDP, the intracellular ROS levels were determined after treatment with AKR1C1 or 1C3 inhibitors. As shown in Figure 9C, when KATO/DDP cells were treated together with cisplatin and ASP9521 or 5PBSA, there was a significant increase in ROS generation after 1 h of cisplatin treatment. Alternatively, KATO/DDP cells could be treated with cisplatin and ASP9521 or 5PBSA with or without NAC. As shown in Figure 9D, cell viability dramatically increased from 48 ± 2.1% to 65 ± 0.69% in cells treated with NAC together with cisplatin and ASP9521 when compared to the combination treatment of cisplatin and ASP952. A similar pattern was observed in NAC treated with cisplatin and 5PBSA at 65 ± 2.8%. 

These outcomes indicated that cisplatin could induce the cell death in KATO cells by generation of intracellular ROS. Moreover, AKR1C1 and AKR1C3 overexpressing KATO/DDP showed a reduction in the generation of intracellular ROS, and while inhibition of these enzymes could regenerate intracellular ROS, this in turn promoted cell death and reversed the resistance property.

## 3. Discussion

In the clinical management of gastric cancer, cisplatin combination regimes have been used as a first-line chemotherapeutic agent [15], while there has been an emergence in drug resistance even with the best treatment. Much effort has been put into understanding the mechanisms of cisplatin chemoresistance; however, the underlying mechanisms are not fully understood, especially regarding SRCGC. In our previous study, we developed a model to acquire cisplatin resistance in KATOIII signet ring cell gastric carcinoma patients by stepwise treatment with cisplatin for a period of 9 months [16]. Again, in this study we found that the resistance index of KATO/DDP exhibited a 7-fold change, which confirmed that this cell presented a significantly greater tolerance to higher cisplatin concentrations when compared to the parental counterparts. Next, we used transcriptomic sequencing strategy to identify the DEGs and to conduct in-depth studies on the mechanisms of cisplatin resistance in SRCGC.

The RNA-seq analysis showed 519 differentially expressed genes between KATOIII and KATO/DDP. Among the topmost significant pathways in the David functional annotation biological process, five pathways, namely oxidation reduction, digestion, steroid metabolic process, and daxorubincin and daunorubincin metabolism, showed the presence of AKR1C1 and AKR1C3 genes. Moreover, AKR genes were also involved in the top three pathways associated with Gene Set Enrichment Analysis, including the oxidation reduction pathway, the small molecule metabolic process, and the cofactor metabolic process. We also constructed the protein–protein interaction network, which identified 13 genes (*UGT1A1, UGT1A10, CYP1A1, CBR3, HSD17B2, AKR1C1, AKR1C3, CXCL8, CXCL11, CXCL13, NMUR2, ADCY7,* and *SSTR5*) in the strongest interacting network. Our MCODE results correlated with those of a study conducted by Ebert et al. that indicated that the activation of phase I metabolism by CYPs generated metabolites and ROS. These metabolites and ROS were capable of inducing Nrf2-regulated antioxidant response genes such as *CBR3* and *AKRs* [17]. Furthermore, *CYP1A1, AKR1C1, AKR1C3,* and *CBR3* were contained in the most significant module analyzed by MCODE. All these combined bioinformatic results indicated that the *AKR1C1* and *AKR1C3* genes are promising molecules in the development of cisplatin resistance in SRCGC. 

Previous studies have reported that the drug transporter genes *MRP1* and *MDR1* mediated the development of platinum drug resistance in lung cancer cells [18,19]. Moreover, nucleotide excision repair and base excision repair mechanisms are also said to be able to protect against the cytotoxicity of cisplatin in cancer cells [20]. Accordingly, the expression levels of *MRP1, MDR1, ERCC1,* and *XRCC1* genes have been determined. No significant fold-change differences were observed between KATOIII and KATO/DDP. High expression levels of AKR family members have been reported to be able to promote drug resistance in cancer cells. Here, we found that AKR1C3 and AKR1C1 exhibited the highest degree of fold-change differences among the AKR family, while AKR1C3 exhibited the highest expression in KATO/DDP. Our Western blot results also supported the finding that AKR1C1 and AKR1C3 were overexpressed in KATO/DDP when compared to KATOIII. These results suggested that AKR1C1 and AKR1C3 are the most important molecules in cisplatin resistance. This determination was supported by the outcomes of previous studies, which found that AKR1C1 and AKR1C3 could catalyze the inactivation of doxorubicin cytotoxicity and the induction of both enzymes to significantly mitigate the cytotoxicity of danorubicin in leukemic U937 cells [8]. Moreover, the involvement of upregulated AKR1C1 and 1C3 in oxaliplatin-resistant gastric cancer cells has been reported [21].

Members of the AKR family could metabolize certain endogenous substrates, such as prostaglandins, steroids, and xenobiotics, in a NADPH-dependent manner. Members of the AKR1C subfamily are emerging as important mediators of cancer pathogenesis. Among them, AKR1C1 and AKR1C3 have been reported to be upregulated in human tumors and identified as prognostic markers for various forms of cancer, including breast, colon, bladder, and prostate cancers [22,23,24]. In the study of colon cancer, specific inhibitors of AKR1C1 and AKR1C3 or knockdowns of the genes in the resistant cells were used to resensitize the cells to cisplatin toxicity [25]. Accordingly, these outcomes were consistent with our results, which found that the inhibition of AKR1C3 or AKR1C1 activity could enhance the cytotoxicity of cisplatin in KATO/DDP cells. This would suggest that AKR1C1 and AKR1C3 are promising factors in promoting cisplatin resistance in SRCGC.

The antitumor activity of cisplatin is a complex process in which several pathways are involved, leading to cell cycle arrest, apoptosis, ferroptosis, and autophagy, depending on the treatment conditions and the cell type [26,27,28]. Cisplatin-induced cell death via apoptosis is the predominant pathway. The previous study indicated that knockdown AKR1C3 in esophageal adenocarcinoma cells exhibited greater apoptosis upon receiving cisplatin treatment. Therefore, we investigated whether the enhancement activity of AKR1C1 and 1C3 inhibitors on cisplatin-induced cell death was associated with apoptosis. Surprisingly, we found no significant differences in percent cell apoptosis between treatments involving cisplatin alone and cisplatin combined with ASP9521 or 5PBSA. Moreover, KATO/DDP cells treated with cisplatin alone induced the expression of cleaved caspase-3. However, a combination treatment did not change the levels of cleaved caspase-3 and Bax when compared with cisplatin alone. The data suggested that cell death induced by AKR1C1 or AKR1C3 inhibitors was independent of the apoptosis pathway. Based on previous studies, AKR1C activity was inhibited along with the expression of sensitized cancer cells to chemotherapeutic drugs via the autophagy pathway [29,30]. Thus, AKR1C1 and 1C3 inhibitors were used to examine whether they could enhance cisplatin-induced KATO/DDP cell death via autophagy. The expression level of LC3BII, an autophagy marker, showed increased expression levels in cisplatin treatment in conjunction with AKR1C1 and 1C3 inhibitors when compared to cisplatin alone. A previous study indicated that mono platinum could induce cell death in human ovarian carcinoma by activating autophagic cell death [31], which was consistent with our results. This outcome was confirmed through the use of MDC dye for the purposes of autophagy vacuole staining. It was found that an increase in the formation of autophagic vacuoles was detected in cisplatin and the inhibitor combination group when compared to cisplatin alone. To confirm that autophagy is a major process of AKR1C1 and 1C3 inhibitors in the enhancement of cisplatin-induced KATO/DDP cell death, we used 3MA as an autophagy inhibitor. The obtained results indicated that when KATO/DDP cells were treated together with cisplatin, ASP9521 or 5PBSA, and 3MA, a reversal was observed in the cytotoxicity of the combination treatment of cisplatin and AKR1C1 or the 1C3 inhibitors. This result correlated with the formation of autophagic vacuoles. In experiments in which cisplatin was combined with AKR1C inhibitors and 3MA, there was a significant reduction in the formation of autophagic vacuoles when compared to cisplatin treated in conjunction with AKR1C inhibitors. 

Cisplatin is one of the most frequently used cytotoxic agents in the treatment of gastric cancer. The traditional mechanism of cisplatin involves the formation of inter- and intrastrand chain cross-linking of DNA for the induction of p53, cell cycle arrest, and apoptosis. More recently, it has been shown that ROS generated by cisplatin could increase lipid peroxidation [32]. The metabolized products of lipid peroxidation can be decomposed to yield a wide range of cytotoxic products, most of which are known to be aldehydes such as malonaldehyde, hexanal, and 4-HNE [33,34]. The AKR1C family can reduce 4-HNE to the nontoxic 1,2-dihydroxynonene, which would play an important role in the detoxication of this reactive aldehyde [33,35]. In the previous study, it was reported that the AKR1C family was involved in oxaliplatin resistance by neutralizing ROS generated by oxaliplatin [21]. Moreover, the inhibition of AKR1C3 expression has been reported to promote an increase in ROS and a reversal of drug resistance to cisplatin in patients diagnosed with colon cancer [25]. Therefore, we determined the appropriate levels of ROS in both KATOIII and KATO/DDP cell lines. The ROS levels increased significantly after cisplatin treatment in KATOIII cells, but not to a significant degree in the AKR1C1 and 1C3 overexpression of KATO/DDP cells. Upon inhibition of AKR1C1 and 1C3 in KATO/DDP cells, ROS increased to a significant level after cisplatin treatment. This indicates that cisplatin could induce ROS generation in KATO cells in a way that was dependent upon the activity of AKR1C1 and 1C3. Nrf2 is a transcriptional factor that is activated by modification of Cys thiols in Keap1 in response to activators such as ROS, electrophiles, and nitrogen radicals [28,36]. Moreover, a previously published report stated that AKR1C3 is said to be a direct target of Nrf2 and plays an essential role in redox balance. To confirm that ROS generation could induce cell death in KATO cells, KATOIII cells were treated in conjunction with cisplatin and NAC. This resulted in an increase in cell survival when compared to treatments with cisplatin alone. 

Furthermore, the inhibition of ROS generation by NAC in cisplatin combined with the ASP9521 or 5PBSA group could significantly mitigate the cytotoxicity of cisplatin to increase KATO/DDP cell survival. According to previous reports, intracellular ROS may enhance ER stress and promote autophagic cell death under certain conditions [37]. In a study conducted by Chen et al., induction of oxidative stress in the U87 and HeLa cell lines promoted autophagic cell death independently of apoptosis, while blocking ROS generation to effectively reverse cell death induced by autophagy [38]. Moreover, curcumin treatment increased intracellular ROS levels in human colon cancer cells and enhanced nonapoptotic cell death through the upregulation of LCIIIB [39]. These data may support our finding that inhibition of AKR1C1 and AKR1C3 may abolish the ROS neutralizing action by aldoketoreductase-overexpressing cells, restore the accumulation of ROS, and promote cell death via autophagy in cisplatin resistance in SRCGC.

## 4. Materials and Methods

### 4.1. Chemicals and Reagents 

Dulbecco’s Modified Eagle Medium (DMEM), trypsin, and penicillin–streptomycin were supplied from Gibco (Grand Island, NY, USA). Fetal bovine serum (FBS), RIPA buffer, protease inhibitors, and Coomassie Plus™ Protein Assay Reagent were obtained from Thermo Scientific Company (Waltham, MA, USA). Apoptosis annexin V and PI was purchased from Biolegand (San Diego, CA, USA). Nucleozol RT reagent was purchased from Takara Bio (Mountain View, CA, USA). Maxima SYBER Green qPCR Master Mix and RevertAid First Strand cDNA Synthesis Kit were purchased from Thermo Scientific (Waltham, MA, USA). Acryalmide solution was purchased from Himedia (L.B.S.Marg, Mumbai, India). Antibodies specific to AKR1C1, AKR1C3, caspase-3, Bax, and LC3B were purchased from Abclonal (Woburn, MA, USA). Nitrocellulose membrane and ECL reagent were supplied by GE Healthcare (Little Chalfont, UK). The 3MA, DCF-DA, and cisplatin were obtained from Sigma (St. Louis, MO, USA).

### 4.2. Cell Line and Culture Condition

The KATOIII human gastric cancer cell line was purchased from American Type Culture Collection (Manassas, VA, USA). The cells were cultured in DMEM containing 10% FBS and 1% (*v*/*v*) penicillin and streptomycin. These cultured cells were maintained at 37 °C in a 95% humidified atmosphere and 5% CO_2_ conditions. 

### 4.3. Establishment of Cisplatin Resistance Gastric Cancer Cell Line

In order to establish a cisplatin-resistant gastric cancer cell line with the same genetic background, the parental KATOIII gastric cancer cell lines were stepwise treated with cisplatin concentration from 0.5 to 3  µM over 9 months, and this cell line was termed KATO/DDP [16]. The resistance ratio was determined by dividing the IC_50_ of the resistance cell line to the IC_50_ of the sensitive ones. Then, the resistance phenotype was maintained by 3 µM concentration of cisplatin, and these cells were cultured in drug-free media for at least two passages before each experiment.

### 4.4. Cell Viability Test

Gastric cancer cells KATOIII and KATO/DDP cells (1.2 × 10^5^) were plated in 12^_^well plates and treated with different concentrations of cisplatin (0–100 µM) or AKR1C inhibitors or 3MA or NAC at 37 °C for 48 h. After 48 h, the cells were collected and resuspended in incomplete media and stained with 0.4% trypan blue dye. The cell viability was assessed by trypan blue staining, and the percentage of cell viability was calculated and compared to that of the control. The IC_50_ value was defined as the required concentration of cisplatin that was required for 50% cell death.

### 4.5. RNA Isolation, Transcriptomic Sequencing, and Analysis

Total RNA was extracted by Nucleozol RT reagent according to the manufacturer’s instructions, and RNA degradation and contamination was checked by Clorox-denatured 1% agarose gel electrophoresis. RNA sequencing was conducted by Novogene Co., Ltd. using the illumina Hiseq 2000 platform. The quality control (QC) of raw reads was checked, and the raw fastq data were analyzed by using CLC genomic work bench version 21. The differentially expressed genes were also analyzed with the CLC Genomics Workbench plugin.

Overexpressed genes were validated using qPCR and Western blot analyses comparing parental and cisplatin-resistant cell lines. The sequencing data were submitted to the National Center for Biotechnology Information (NCBI) Sequence Read Archive (SRA) under accession number GSE186205. For gene functional annotation, the differentially expressed genes were analyzed by using Gene Set Enrichment Analysis version 4.1.0, David Bioinformatics Resource functional annotation tools version 6.8, and Cytoscape version 3.8.2.

### 4.6. RNA Isolation, Reverse Transcription, and Quantitative PCR

Total RNA was extracted by Nucleozol RT reagent according to the manufacturer’s instructions. Complementary DNA was synthesized by using the Revert Aid First Strand cDNA Synthesis Kit (Thermo Scientific, Loughborough, UK). The quantitative reverse transcription polymerase chain reaction was carried out by Maxima SYBER Green qPCR Master Mix (Thermo Scientific, Loughborough, UK) with the applied biosystem 7500 Fast platform. The relative expression of the genes was analyzed by 2^−ΔΔCT^. The primer sequences used in this study are listed in Appendix A. 

### 4.7. Western Blotting Analysis 

The treated cells were extracted by a RIPA lysis buffer containing protease inhibitors (1 mM PMSF, 10 μg/mL leupeptin, 10 μg/mL aprotinin) for 20 min on ice. The insoluble matter was removed by centrifugation at 12,000 rpm for 15 min at 4 °C, the supernatant fraction was collected, and the protein concentration was determined by using a Bradford protein assay. Total protein extracted from the cells was separated by 10% and 12% SDS polyacrylamide gel electrophoresis and then transferred to the nitrocellulose membrane. The membrane was then incubated with the desired primary antibody overnight at 4 °C. The secondary antibody at 1:10,000 was incubated for 2 h at room temperature, and the results were visualized by a chemiluminescent detection system and then exposed to X-ray film (GE Healthcare Ltd., Little Chalfont, UK).

### 4.8. Intracellular ROS Measurement

Intracellular ROS was measured by a DCF-DA (2′,7′-dichlorofluorescin diacetate) assay. A total of 4 × 10^5^ cells were treated with various concentration of cisplatin with or without AKR1C1 and 1C3 inhibitors for 1 h and stained with a 5 µM final concentration of DCF-DA for 30 min at 37 °C. The cells were washed with PBS and then lysed by 90% Dimethyl sulfoxide, and the fluorescence intensity was measured against 485/525 by fluorescent plate reader.

### 4.9. Apoptosis Assay

KATO/DDP cells (2 × 10^5^) were treated with cisplatin with or without AKR1C1 and 1C3 inhibitors and incubated at 37 °C for 24 h. After 24 h, the cells were collected and washed with PBS and then stained with annexin V and PI for 30 min at 37 °C, and analyzed with the BD FACScan^TM^ flow cytometer (BD Biosciences, San Jose, CA, USA).

### 4.10. Monodensylcadaverine Staining

The KATO/DDP cells (2 × 10^5^) were treated with cisplatin with or without AKR1C1 and 1C3 inhibitors and 3MA and incubated for 24 h. Then, the cells were washed with ice-cold PBS, and stained by MDC dye for 30 min at 37 °C. The cells were washed with PBS twice to remove the excess MDC. The cells were then visualized with a fluorescence microscope (Carl Zeiss AG, Jena, Germany) at an excitation wavelength of 460–500 nm and an emission wavelength of 512–542 nm.

### 4.11. Statistical Analysis

Each of the experiments was conducted in triplicate. Statistical analysis was performed using SPSS (version 22). Data were expressed as mean ± SD with the number of individual experiments described in the figure legends. For comparisons between the means of two variables, unpaired student’s *t*-test was used. For comparisons among multiple variables, one-way ANOVA was used. The statistical significance of differences was considered significant when the *p*-value was <0.05.

## 5. Conclusions

Our research determined that AKR1C1 and AKR1C3 could play a dominant role in promoting drug resistance by neutralizing the ROS generated by cisplatin. Meanwhile, the inhibition of AKR1C3 and 1C1 effectively upregulated ROS generation, increased the cytotoxicity of cisplatin, and promoted autophagic cell death, while reversing the cisplatin resistance property in signet ring gastric carcinoma patients as a unique biological behavior.

## Figures and Tables

**Figure 1 ijms-22-12512-f001:**
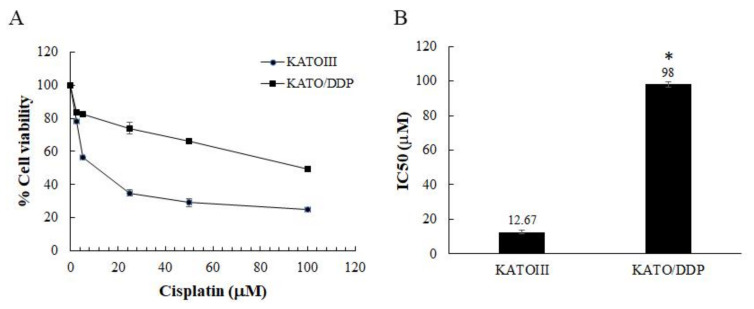
Response of parental drug-sensitive KATOIII and drug-resistant KATO/DDP cells to cisplatin cytotoxicity. KATOIII and KATO/DDP cells were treated with different concentrations of cisplatin (0–100 µM) for 48 h, and cell viability was assessed by trypan blue cell exclusion assay. (**A**) Cell viability of KATOIII and KATO/DDP. (**B**) IC _50_ value of cisplatin in KATOIII and KATO/DDP cell line. The data are presented as means; * represents *p* < 0.05 when compared to control.

**Figure 2 ijms-22-12512-f002:**
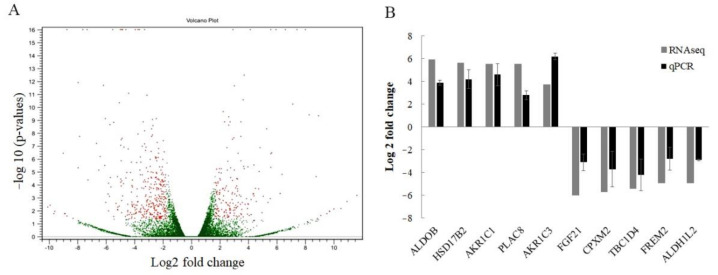
Identification of differentially expressed genes between KATOIII and KATO/DDP cells. (**A**) Volcano diagram of differentially expressed genes between KATO/DDP and KATOIII cell lines, showing 174 upregulated and 345 downregulated genes. (**B**) To validate the differentially expressed genes from RNA sequencing, the expression level of five upregulated and downregulated genes were examined by using qPCR.

**Figure 3 ijms-22-12512-f003:**
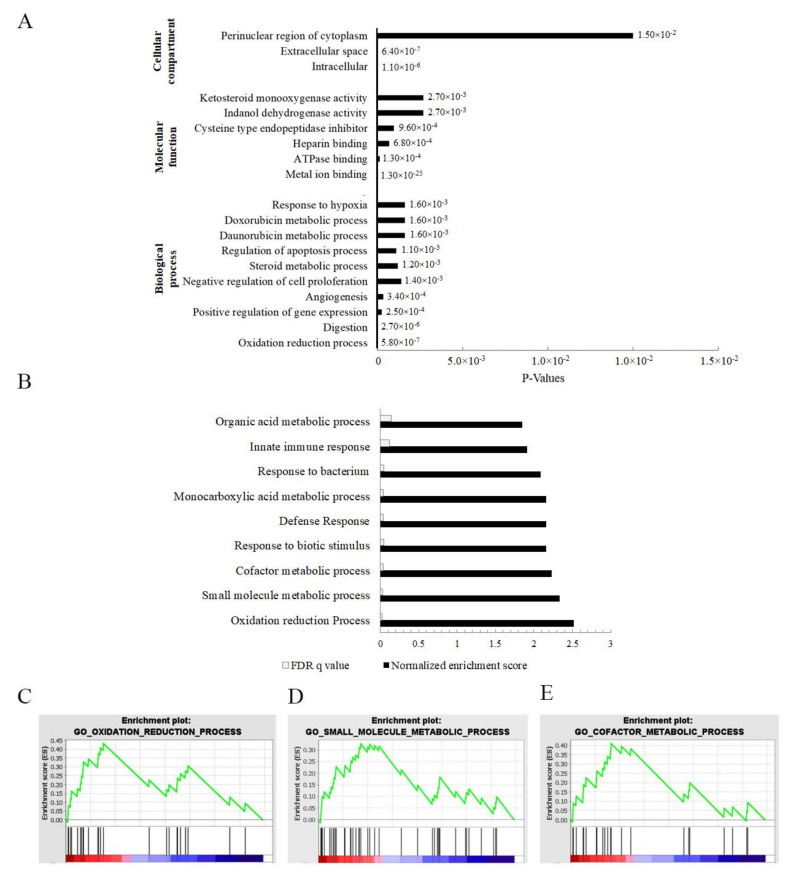
Identification of cisplatin-resistance-related genes in signet ring gastric cancer by bioinformatics analysis of RNA sequencing data: (**A**) David Bioinformatics functional annotation according to gene ontology (GO) in biological process, molecular function, and cellular compartment; (**B**) top 10 GO terms of Gene Set Enrichment Analysis; (**C**) oxidation reduction GO term of Gene Set Enrichment Analysis; (**D**) small molecule metabolic process GO term of Gene Set Enrichment Analysis; (**E**) cofactor metabolic process GO term of Gene Set Enrichment Analysis.

**Figure 4 ijms-22-12512-f004:**
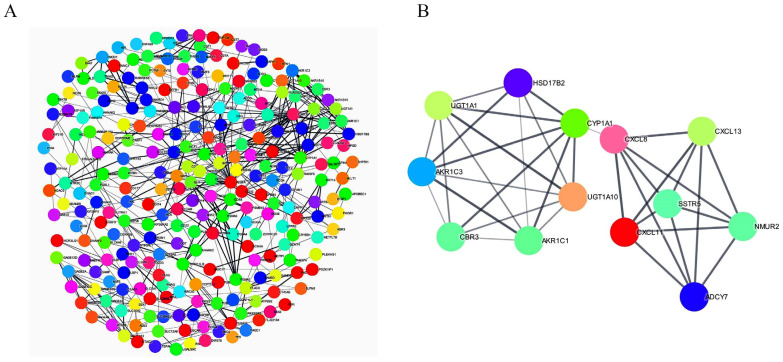
Protein–protein interaction network construction. (**A**) Protein–protein interaction network of differentially expressed genes, constructed using Cytoscape, showing 371 nodes with 483 edges. (**B**) Topmost significant module in PPI network analyzed by MCODE (score = 6).

**Figure 5 ijms-22-12512-f005:**
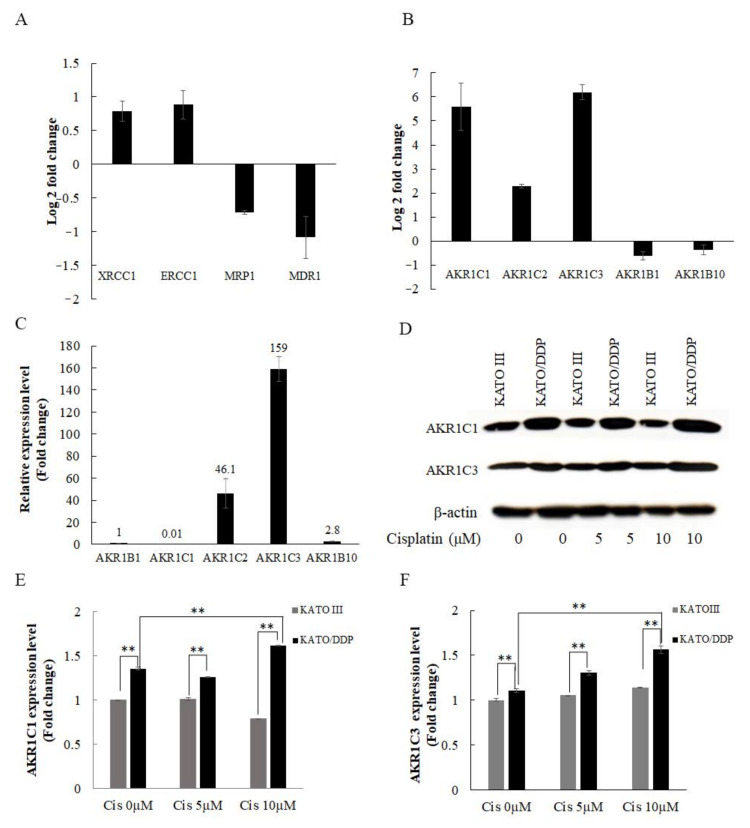
Determination of cisplatin resistance related genes in SRCGC. (**A**) Fold-change difference in expression of known drug-resistance-related genes between KATO/DDP and KATOIII gastric cancer cell lines by qPCR. (**B**) Fold-change difference in expression of aldoketoreductase family between KATO/DDP and KATOIII gastric cancer cell line by qPCR. (**C**) Relative expression of aldoketoreductase family in KATO/DDP cell line. (**D**) KATO/DDP and KATOIII gastric cancer cell lines were treated with different concentration of cisplatin (0–10 µM) for 24 h. The expression levels of AKR1C1 and AKR1C3 were detected by Western Blot analysis. (**E**) Quantitative representation of AKR1C1 protein expression by Western blot. (**F**) Quantitative representation of AKR1C3 protein expression by Western blot analysis. Data are presented in mean ± SD; ** represents *p* < 0.01.

**Figure 6 ijms-22-12512-f006:**
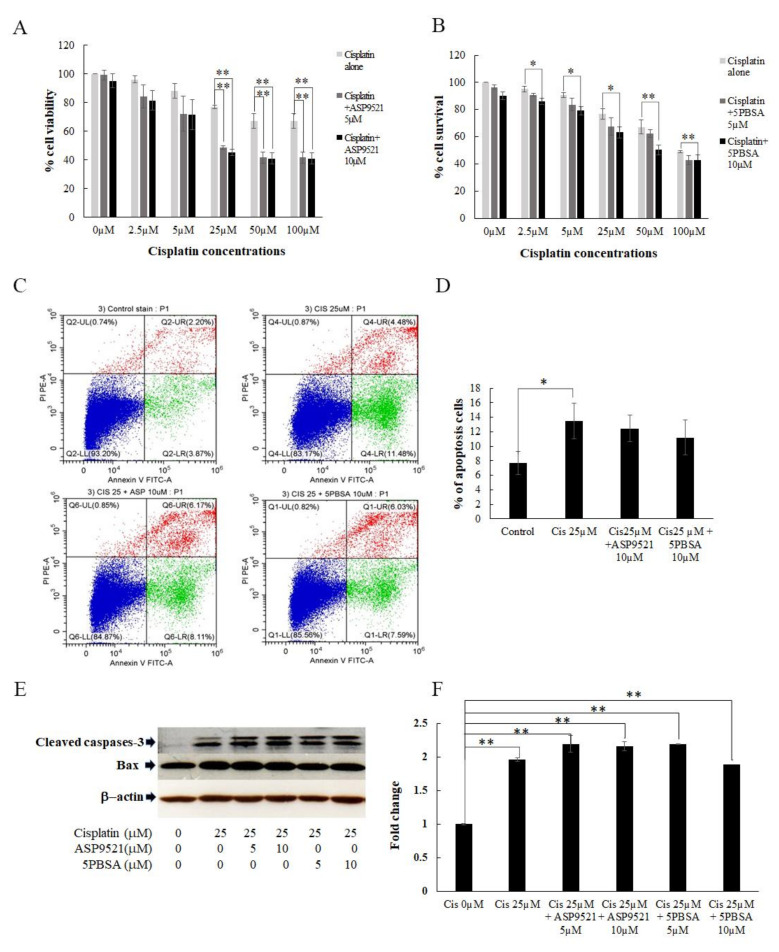
AKR1C1 and AKR1C3 regulated cisplatin-induced KATO/DDP cell death. (**A**) Cell viability of KATO/DDP cells treated with cisplatin (0–100 µM) with or without ASP9521 at 5 µM and 10 µM by trypan blue cell exclusion assay. (**B**) Cell viability of KATO/DDP cells treated with cisplatin with or without 5PBSA at 5 µM and 10 µM by trypan blue cell exclusion assay. (**C**) KATO/DDP cells were treated with cisplatin with or without ASP9521 or 5PBSA at 10 µM for 24 h, and we analyzed the apoptosis assay by flow cytometer. (**D**) Quantitative representation of % of apoptosis analyzed by flow cytometer. (**E**) KATO/DDP cells were treated with cisplatin with or without ASP9521 or 5PBSA at 10 µM for 24 h, and apoptosis markers were detected by Western blot analysis. (**F**) Quantitative representation of apoptosis marker cleaved caspase 3 by Western blot analysis. Data are presented in mean ± SD; * represents *p* < 0.05 and ** represents *p* < 0.01.

**Figure 7 ijms-22-12512-f007:**
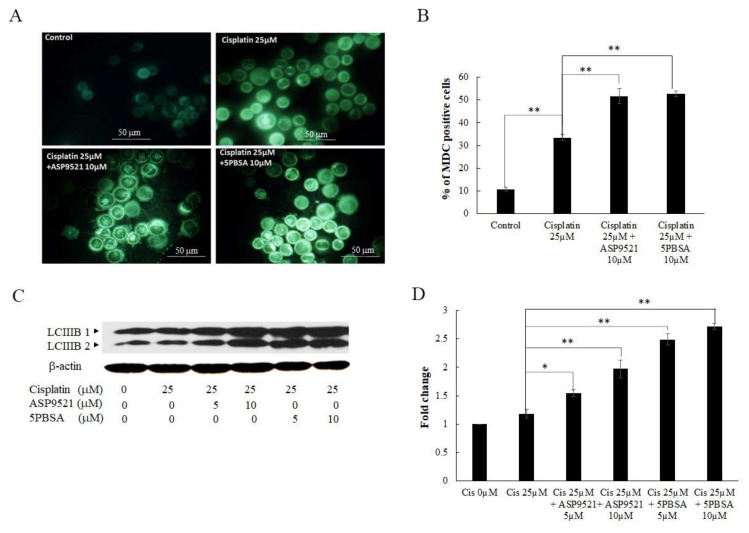
Inhibition of AKR1C1 and AKR1C3 induced cell death via autophagy. KATO/DDP cells were treated with cisplatin with or without ASP9521 or 5PBSA at 10 µM for 24 h. (**A**) Autophagic vacuoles were stained with MDC dye and visualized under a fluorescent microscope. (**B**) Graphical representation of % of MDC-positive cells in KATO/DDP cells. (**C**) Western blot analysis of autophagy marker. (**D**) Quantitative representation of autophagy marker by Western blot analysis. Data are presented as mean ± SD; * represents *p* < 0.05 and ** represents *p* < 0.01.

**Figure 8 ijms-22-12512-f008:**
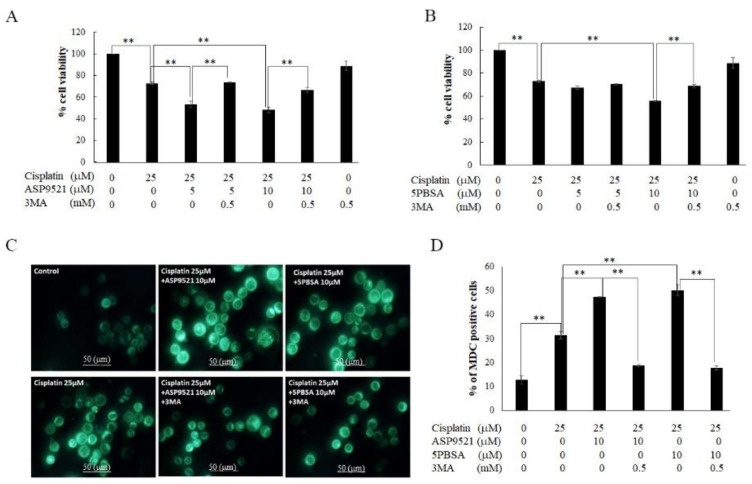
Verification of autophagy as a major pathway of AKR1C1- and 1C3-regulated cell death. (**A**) KATO/DDP cells were treated together with cisplatin and ASP9521 with or without 3MA at 0.5 mM. Cell viability was assessed by using trypan blue cell exclusion assay. (**B**) KATO/DDP cells were treated together with cisplatin and 5PBSA with or without 3MA at 0.5 mM, and cell viability was assessed by using trypan blue cell exclusion assay. (**C**) KATO/DDP cells were treated together with cisplatin and ASP9521 or 5PBSA at 10 µM with or without 3MA at 0.5 mM. Autophagic vacuoles were stained with MDC dye and visualized under fluorescent microscope. (**D**) Graphical representation of % of MDC-positive cells in KATO/DDP. Data are presented as mean ± SD; ** represents *p* < 0.01.

**Figure 9 ijms-22-12512-f009:**
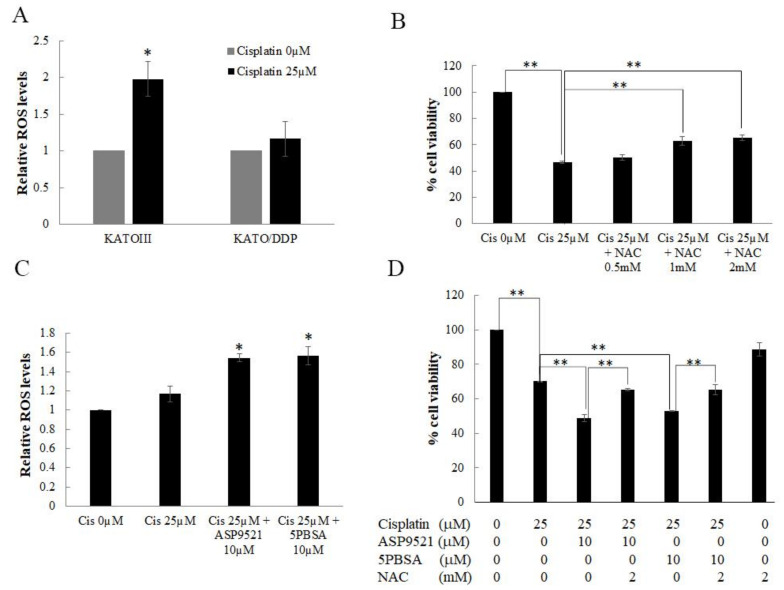
Role of intracellular ROS in AKR1C1 and 1C3 mediating chemo-resistance in signet ring gastric cancer cells. (**A**) Intracellular ROS level in KATOIII and KATO/DDP cell lines after 1 h treated with cisplatin was detected by DCF-DA ROS assay. (**B**) KATOIII cells were treated with cisplatin with or without NAC for 48 h at 37 °C. Cell viability was assessed by using trypan blue cell exclusion assay. (**C**) Intracellular ROS level in KATO/DDP cells after treatment with cisplatin and ASP9521 or 5PBSA. KATO/DDP cells were pretreated with ASP9521 or 5PBSA at 10 µM for 3 h and added with cisplatin at 25 µM. Intracellular ROS was measured by DCF-DA ROS assay after 1 h of cisplatin treatment. (**D**) Cell viability of KATO/DDP cells treated with cisplatin and ASP9521 or 5PBSA with or without NAC. Data are presented as mean ± SD; * represents *p* < 0.05 and ** represents *p* < 0.01.

## Data Availability

The sequencing data were submitted to the National Center for Biotechnology Information (NCBI) Sequence Read Archive (SRA) under accession number GSE186205.

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
