# Peer review of "Transcriptomic Profiling Reveals AKR1C1 and AKR1C3 Mediate Cisplatin Resistance in Signet Ring Cell Gastric Carcinoma via Autophagic Cell Death"

_ijms, 2021, doi:10.3390/ijms222212512_

Round 1
Reviewer 1 Report
In this manuscript, the authors provide interesting and important information on the mechanism of resistance to cisplatin treatment in gastric carcinoma cells. Two essential genes AKR1C1 and AKR1C3 were identified as a possible reason for the cisplatin resistance was shown that inhibition of this gene will lead to sensitivity to cisplatin. This manuscript can be accepted after minor revision. It will be interesting to see how co-administration with ROS promoter will change resistance to the cisplatin/
Author Response
Reviewer 1 General comment: “It will be interesting to see how co-administration with ROS promoter will change resistance to the cisplatin”
Response: We have edited in discussion section line 427- 431 describing the role of ROS promoter and cisplatin resistance.
Reviewer 2 Report
Phoo et al. reported interesting findings regarding cisplatin resistance in signet ring cell gastric carcinoma. The study is comprehensive and informative, yet the data presentation and explanation could be improved to clarify. Specific comments are listed below:
- Figure layout should aim for clarity. For example, the font sizes should be more readily readable on an A4 printout. Resolutions should be consistent without distortion among the panels. All figures could be benefited from such improvements to help the reader SEE the data and annotations. All sub-panels should share the same font style and sizes. For example, Figure 3 f-g can never be understood and read by anyone. Figure 3a-b and nearly all bar charts are made with heavy distortion. Scale bars in all figures are barely visible and readable.
- The author should also try their best to illustrate the data while explaining the data. All figure captions lack brief summaries for each panel, on top of not having an overall summary per figure.
- Western blotting data is confusing. The relative expression normalized to AKR1B1 as 1-fold should be placed on the left (for example Fig. 4C. This is done right in Figure 4F). High and low-dose Cisplatin at 5 and 10 uM is a continuous variable instead of a categorical variable; + - annotation in Figure 4C does not make sense. The same applied to other dose-related data representations. Most WB images are over-exposed for proper quantification, such as Figure 6C.
- The author should include an abstract figure like an illustration to better represent the biology involved for a broader readership. At the moment, the manuscript reads very much like talking to the lab insider-only with almost no effort to build or guild the readers with proper background knowledge (almost like reading lab book figures).
Author Response
General comment: “the data presentation and explanation could be improved to clarify”
Response: Per advice, we revised the introduction section in page 2, line 45-54 and 62-66. Furthermore, we have improved the way of data presentation and explanation in the result section as well. We have also edited in result section 2.2, page 4, line 132-133. In result section 2.3, page 6, line 171-173 by adding some explanation for selecting AKR1C1 and AKR1C3 as cisplatin resistance promoting genes. We also edited the precise conclusion in result section 2.4, page 8, line 212-218. In result section 2.5, page 11, line 262-265 and result section 2.6, page 12, line 303-307, we replaced with better explanatory phrases to point out the purpose of that experiment.
Specific comment
Comment 1: Figure layout should aim for clarity. For example, the font sizes should be more readily readable on an A4 printout. Resolutions should be consistent without distortion among the panels. All figures could be benefited from such improvements to help the reader SEE the data and annotations. All sub-panels should share the same font style and sizes. For example, Figure 3 f-g can never be understood and read by anyone. Figure 3a-b and nearly all bar charts are made with heavy distortion. Scale bars in all figures are barely visible and readable.
Response: Per the reviewer’s advice, all of the figures were modified to be readily readable with consistent resolution according to reviewer suggestion. We replaced the font with steady Time New Roman with size 12-14 in almost all figures. Due to the problem of readable size and clarity of the gene list in figure 3C-E, we shifted them to Table S5. Moreover, Figure 3 F and G were separated as Figure 4A and B with higher quality which can assess clearly in online format.
The rest figures’ number was changed according to order. To provide a concise information, and better quality image, previously described figure 6 (now in order Figure 7) was subdivided into figure 7 and 8.
Comment 2: The author should also try their best to illustrate the data while explaining the data. All figure captions lack brief summaries for each panel, on top of not having an overall summary per figure.
Response: We’ve changed the caption of Figure 1 in page 3 line 100. In Figure2, we have edited the legend of figure 2B in page 4, line 145-146. We updated the Figure 3 caption in page 5, line 148-149 and also figure 4 in page 6, line 155-157. We modified the legend of Figure 5 E and F in page 7, line 189-190. We edited the caption of Figure 6 in page 9, line 220. As we separated Figure 7 into Figure 7 and 8, we updated the caption of Figure 8 in page 11, line 267. We also revised the caption of Figure 9 and legend of Figure 9A and D in page 13, line 309-311 and 314-315.
Comment 3: Western blotting data is confusing. The relative expression normalized to AKR1B1 as 1-fold should be placed on the left (for example Fig. 4C. This is done right in Figure 4F). High and low-dose Cisplatin at 5 and 10 uM is a continuous variable instead of a categorical variable; + - annotation in Figure 4C does not make sense. The same applied to other dose-related data representations.
Response: In Figure 5C (previously Figure 4C), we have modified the illustration of AKR1B1 as 1-fold to the left side of the figure. For high and low dose of cisplatin as a continuous variable, we’ve edited in Figure 5D (previously Figure 4D), Figure 6E (previously Figure 5E), Figure 7C (previously Figure 6C), Figure 8 A, B, D (previously Figure 6 E,F,H) and Figure 9D (previously Figure 7D).
Comment 4: Most WB images are over-exposed for proper quantification, such as Figure 6C.
Response: In Figure 7C (previously Figure 6C), we’ve replaced with new one that has less exposure for proper quantity assessment.
Comment 5: The author should include an abstract figure like an illustration to better represent the biology involved for a broader readership.
Response: We’ve also added the graphical abstract.
All the corrections have been highlighted in yellow color. We hope our revised manuscript are now able to reconsider for publication in Interantional Journal of Molecular Sciences.
Thank you very much for your time and kind assistance.
With my best regards,
Supachai Yodkeeree
Assistant Professor,
Department of Biochemistry, Faculty of Medicine,
Chiang Mai University, Chiang Mai 50200, Thailand